# PCR Mediated Nucleic Acid Molecular Recognition Technology for Detection of Viable and Dead Foodborne Pathogens

**DOI:** 10.3390/foods11172675

**Published:** 2022-09-02

**Authors:** Mengtao Chen, Xinyue Lan, Longjiao Zhu, Ping Ru, Wentao Xu, Haiyan Liu

**Affiliations:** 1Research Center for Sports Nutrition and Eudainomics, Institute for Sports Training Science, Tianjin University of Sport, Tianjin 301617, China; 2School of Public Health, North China University of Science and Technology, Tangshan 063210, China; 3Key Laboratory of Precision Nutrition and Food Quality, Department of Nutrition and Health (Institute of Nutrition and Health), China Agricultural University, Beijing 100193, China

**Keywords:** polymerase chain reaction, viable bacteria detection, dead bacteria detection, DNA-intercalating dyes, food safety, foodborne pathogens

## Abstract

Living foodborne pathogens pose a serious threat to public and population health. To ensure food safety, it is necessary to complete the detection of viable bacteria in a short time (several hours to 1 day). However, the traditional methods by bacterial culture, as the gold standard, are cumbersome and time-consuming. To break through the resultant research bottleneck, PCR mediated nucleic acid molecular recognition technologies, including RNA-based reverse transcriptase PCR (RT-PCR) and DNA-based viability PCR (vPCR) have been developed in recent years. They not only sensitively amplify detection signals and quickly report detection results, but also distinguish viable and dead bacteria. Therefore, this review introduces these PCR-mediated techniques independent of culture for viable and dead foodborne pathogen detection from the nucleic acid molecular recognition principal level and describes their whole-process applications in food quality supervision, which provides a useful reference for the development of detection of foodborne pathogens in the future.

## 1. Introduction

Due to the characteristics of small size and strong reproductive capacity, bacteria exist in food, feed, drinking water and almost everywhere near human life. In addition to spores, there are three living states of foodborne pathogens: viable, dead, and viable but nonculturable (VBNC) cells [1]. Nevertheless, only living bacteria can pollute food or the environment, and infect livestock and crop products. Therefore, in the quantitative detection of living bacteria, the ideal situation is to eliminate the interference of dead bacteria from the matrix.

The traditional microbial detection method needs to culture the foodborne pathogens in the sample for a long time according to standard steps, and then count the colonies on the medium. Although long-term practice has proved that the traditional culture method is considered as the gold standard [2], there are still inherent challenges of time-consuming and laborious requirements. Polymerase chain reaction (PCR) is a highly sensitive and specific molecular biology technology [3], which is widely used in the detection of various pathogens. Based on the principle of DNA double-strand replication to amplify specific DNA fragments outside the organism, PCR realizes the efficient detection of target bacteria. However, DNA in dead organisms may also be intact, which limits the accuracy of PCR in distinguishing living and dead bacteria in the samples. Accordingly, high-quality programs are urgent needed to fill the gaps in the detection of viable bacteria.

In recent years, several kinds of molecular methods based on PCR for live bacteria detection have been developed, such as molecular activity test (MVT) and viability PCR (vPCR). According to the principle that only viable cells can actively transcribe RNA [4], MVT determines whether viable bacteria exist in the sample by measuring the changes in the synthesis of precursor ribosomal RNA. With the assistance of nucleic acid intercalation dyes, DNA amplification-based vPCR has successfully distinguished the physiological state of foodborne pathogens, which is widely used in the detection of various bacteria [5]. In summary, based on the principle of PCR technology for detecting of live bacteria, this paper expounds the advantages and disadvantages of RT-PCR and vPCR, and describes the applications of the two techniques in the detection of foodborne pathogens in the food chain, which provides a useful reference for the future application of molecular amplification technology to detect live bacteria.

## 2. RNA-Based RT-PCR Detecting Technique

The traditional method of detecting foodborne pathogens is almost entirely dependent on the isolation and culture of specific microorganisms from food, and then performing a series of biochemical and serological testing. It takes about 5–10 days to identify whether there are viable bacteria in the sample, which is time-consuming and laborious. Although PCR detection is rapid and specific, it cannot effectively distinguish the bacteria with different physiological states. Because mRNA only exists in viable bacteria, it can act as a reverse transcriptase PCR (RT-PCR) target, which breaks through the bottleneck of rapid identification and detection of viable bacteria (Figure 1A). Jou et al. [6] selected the mRNA of 85B secreted by *Mycobacterium tuberculosis* as an amplification target and used RT-PCR to quantitatively detect viable bacteria. The sensitivity of RT-PCR (12 CFU/mL) was like that of traditional fluorochrome staining (9 CFU/mL), but the former exhibited shorter detection time, fewer detection steps and easier operation, which saved a lot of manpower, material resources and time. Aarthi et al. [7] determined the survival status of bacteria in clinical specimens by RT-PCR to determine a more accurate dosage of antibiotics used in clinical treatment, minimizing the emergence of drug resistant foodborne pathogens. Within 8 h after receiving clinical samples, the method can be used for quantitative detection of live bacteria in any laboratory with basic molecular biology equipment, with a sensitive detection limit (LOD) of 0.4 fg. Their results confirmed that the detection based on RT-PCR bacteria survival state analysis in clinical or environmental samples is feasible.

Compared with the traditional culture method, RT-PCR is more universal and suitable for detecting the active state of extensive microorganisms. However, the final detection results of both schemes are output by agarose gel electrophoresis, which is time-consuming and tedious. In recent years, many research teams have transformed the amplification signal into an electrochemical one, so as to obtain a more intuitive and rapid detection system. Yuan et al. [8] established a cascade signal amplification strategy based on RT-PCR triggering G-quadruplex DNA enzymatic reaction, realizing the visualization and rapid detection of viable *Cronobacter sakazakii* in samples. The RT-PCR amplification products targeting viable bacteria mRNA can be assembled with heme ferrate to generate many G-quadruplex DNAzymes, which further catalyzes H_2_O_2_—mediated TMB oxidation to TMBOX with strong electrochemical activity, resulting in changes in color to blue (Figure 1B). Accordingly, the existence of viable bacteria in the sample can be directly determined by observing the color change of the reagent, and the target can be quantitatively detected by measuring the change of electrochemical signals. The detection range of this new method was 2.4 × 10^7^ CFU/mL–3.84 × 10^4^ CFU/mL, and LOD was 501 CFU/mL. This new electrochemical detection assay is expected to replace electrophoresis analysis and provides a simpler and easier signal output platform for the detection of viable bacteria.

In summary, there are many advantages of RNA-based RT-PCR in the application of detecting viable bacteria, such as low sensitivity, strong specificity, and many emerging portable visualization schemes have been widely established and applied (Table 1). However, the half-life of mRNA is very short (1.5–2 min), and it is a challenge to purify and obtain complete mRNA. Therefore, it is necessary to improve the RT-PCR scheme and optimize the pretreatment steps to develop a larger application platform in viable bacteria detection.

## 3. DNA-Based Viability PCR Detecting Technique

### 3.1. Identification of Live and Dead Bacteria Based on Cell Membrane Integrity

Unlike RT-PCR, vPCR is a promising detection method based on robust DNA. The vPCR technology contains two reaction steps including nucleic acid intercalating dye pretreatment and PCR amplification, which can quickly identify viable bacteria from the dead in the sample. Due to the incomplete cell membrane of damaged or dead bacteria, light-activated nucleic acids embedded in dyes, including ethidium monoazide (EMA) or propidium monoazide (PMA), can penetrate cells to form irreversible binding with DNA molecules, thereby inhibiting subsequent PCR amplification to achieve the differential detection of living and dead bacteria [9]. As vPCR is simple and easy to operate without significantly increasing the detection time, it has been widely used in the identification of live bacteria. For analysis based on DNA templates, Knut Rudi [10] used EMA-PCR to quantitatively analyze the viable *Campylobacter jejuni* in mixed bacterial samples with 4 log10 dynamic range, which breaks through the limitation that a conventional microscope-based BacLight assay has (Figure 2A). The EMA-PCR was able to detect as low as 10 cell/mL viable *Listeria monocytogenes* in pasteurized milk [11]. The EMA could penetrate 10^3^–10^7^ heat-treated *L. monocytogenes* cells within 30 min and only inhibit hly gene are not other to nucleic acid participate in PCR amplification of viable bacteria. Meanwhile, to reduce the interference of potential inhibitors in pasteurized milk on amplification, the research team developed a new method for real-time PCR targeting using EMA-treated long DNA template [12]. The optimized method completely inhibited the DNA amplification of dead *Escherichia coli* and increased the detection signal of viable *E. coli*, which furtherly promoted the development of detection of viable bacteria in conventional milk. Huang et al. [13] established a PMA-coupled multiplex PCR (PMA-mPCR) based method, which quickly and reliably identified five viable foodborne pathogens in fresh juice. The LODs of this method for *Staphylococcus aureus*, *E. coli*, *Shigella*, *K. pneumoniae*, and *P. aeruginosa* in the samples were 100, 1000, 100, 100 and 100 CFU/mL, respectively. Combining PMA with multiplex quantitative real-time PCR, Liang et al. [14] introduced multiplex fluorescence probes into PCR amplification after PMA/pretreatment, achieving simultaneous real-time detection of three live bacteria in food (Figure 2B). The LODs of *Salmonella*, *E. coli*, and *S. aureus* in pure medium were 100, 100 and 10 CFU/mL, respectively, allowing this method to be used in monitoring microbial contamination in drugs and foods.

As the earliest DNA-intercalating dye, EMA is cheaper and more widely used than PMA. At the same time, PMA is more used by researchers to combine with multiplex PCR to achieve high-throughput detection. Overall, EMA-PCR and PMA-PCR have been successfully applied to distinguish living and dead bacteria in various samples such as water, food, air, and clinical specimens. However, for VBNC cells that may exist in UV-treated samples, above two methods may cause increased false positive results that overestimate the number of viable cells in samples. Therefore, it is necessary to further improve the accuracy of EMA-PCR and PMA-PCR in distinguishing and detecting viable bacteria in future applications.

### 3.2. Identification of Living and Dead Bacteria Based on Cell Metabolic Activity

To improve the above defects of high false positive rate of vPCR, two new DNA-intercalating dyes based on metabolic activity have been devised, namely DyeTox13 Green C-2 Azide (DyeTox13) and thiazole orange monoazide (TOMA). The former preferentially binds to nucleic acids with intact-membrane death cells and enzyme-inactivated cells, leading to bright yellow-green fluorescence with diffuse reflection. The dye also binds active viable cells’ DNA and marks intravacuolar structures clearly with red fluorescence. Compared with the results of EMA-PCR and PMA-PCR, Lee and Bae [15] evaluated the ability of DyeTox13 coupled with qPCR (DyeTox13-qPCR) to distinguish between active/inactive Gram-negative bacteria (*P. aeruginosa PAO1*) and Gram-positive bacteria (*Enterococcus faecalis v583*). By comparing the average ΔCt values of PCR reactions treated with the three dyes, there was no significant difference in the detection results, indicating that their ability to identify active bacteria in ordinary samples is equivalent. Li et al. [16] detected viable and dead bacteria in UV disinfected samples using PMA-PCR and DyeTox13-PCR. Different from the results of the PMA test where only “membrane damaged” cells can be distinguished, the DyeTox13 test showed greater ability on “dormant” cells with no metabolic activity in eggshell samples, which was consistent with the results of plate counting, indicating it is more suitable to detect viable bacteria in samples after UV irradiation (Figure 2C). The latter is a novel photoactive dye which can freely enter cells and cross-link with DNA composed of three components: a nucleic acid-intercalating moiety, a cross linkable moiety, and a linker. The ester bond will be hydrolyzed by active esterase in viable cells, and only DNA in dead cells without esterase activity crosslink with TOMA that inhibits the subsequent PCR amplification. The viable *E. coli* at 1000 CFU/mL could be detected by qPCR after incubated with TOMA for 20 min and then exposed to light for 30 min [17] (Figure 2D). The TOMA-PCR not only exclude the interference of dead cells but also works in extreme conditions such as UV disinfection, low temperature, strong radiation, and oligotrophic ones. Feng et al. [18] detected the active *Klebsiella pneumoniae* in the powdered infant formula (PIF) by recombinase-aided amplification (RAA). Under constant temperature of 39 °C, TOMA-RAA completed the detection of viable target bacteria within 40 min, and the LOD was as low as 2.3 × 10^4^ CFU/mL.

The new nucleic acid intercalation dyes, DyeTox13 and TOMA, can correctly distinguish the survival state of bacteria after various sterilization conditions and do not require additional fluorescent dyes or probes, which can achieve the effect of visual detection by itself without additional fluorescent dyes or probes.

In summary, after continuous optimization, the detection characteristics of vPCR is stable and not affected by dead foodborne pathogens, so it can be widely used in the rapid and accurate detection of living bacteria in environmental, food and other types of samples (Table 1).

## 4. Whole-Processes Application of PCR Mediated Nucleic Acid Molecular Recognition Technology in Detecting Foodborne Pathogens

Foodborne pathogens refer to a large group of bacteria that cause diseases or even death of human beings with food as the carrier and are the main threat to human health. Many PCR-mediated nucleic acid molecular recognition strategies have been constructed by researchers to detect the viable foodborne pathogens in food production, processing, storage, transportation, and sales, which provides an effective tool to avoid the outbreak of foodborne diseases (Figure 3).

**(1) Live foodborne pathogens detection in production and living environment.** Because *Staphylococcus aureus* widely exists in nature and can produce enterotoxin and cause food poisoning, Chang and Lin [23] used the vPCR method to detect live bacteria in the air. The average percentage of live bacteria was 12–44%, which provided direction for improving the food processing environment. To reduce the pollution of aquatic products by food-borne pathogens, Yoon et al. [24] established a real-time quantitative RT-PCR strategy to quantitatively detect three live pathogens (*Escherichia coli*, *Vibrio harveyi* and *Enterococcus faecalis*) in aquaculture water. The Pearson correlation coefficient (R) of RT-PCR for all three pathogenic bacteria was 0.907–0.986, *p* < 0.05, indicating of the high accuracy of this method for monitoring bacterial activity.

**(2) Live foodborne pathogens detection in storage.** Vaitilingom et al. [25] developed a method based on RT-PCR to detect various live bacteria in milk after pasteurization, and successfully applied this method to the detection of live bacteria in yogurt and beer. This method is highly sensitive and can detect 5–10 target living bacteria at the concentration of pollutants as low as 10 cells/mL. Castro [26] used vPCR to evaluate the survival status of *Campylobacter* in frozen and refrigerated chicken. The results showed that compared with the untreated samples, the number of positive refrigerated carcasses detected by PMA pretreatment was less (*p* < 0.05), and there was no significant difference in the total number of *Campylobacter* and the number of viable bacteria in refrigerated chicken whether or not treated by PMA. Therefore, the addition of PMA did not change the survival status of bacteria in the samples, and could quickly assess the number of viable bacteria, which had made great contribution to avoiding the spread of *Campylobacter* through frozen chicken that endangers public health.

**(3) Live foodborne pathogens detection in transportation.** With the development of globalization of the food trade, it is essential to quickly complete the detection of live foodborne pathogens in imported and exported food. Immanuel [27] established a vPCR strategy that can quickly distinguish and detect live bacteria in samples without the interference of pollutants in plant quarantine environment. This method was used to monitor the pollution of the leaf and fruit of strawberry caused by plant pathogenic bacteria—living *Xanthomonas*. This method can detect 10^3^–10^8^ CFU/mL live bacteria in the presence of high concentration dead cells (10^6^ CFU/mL), which has created great application value in sterile food transportation.

**(4) Live foodborne pathogens detection in market.** To detect the existence of live *Listeria monocytogenes* in commercial pork, Ye et al. [28] developed a real-time RT-PCR without pre-enrichment step and evaluated its reliability by comparing the detection results with those of traditional culture methods. With the advantages of fast detection speed, strong sensitivity, and wide detection range (10–10^6^ CFU/mL), this method effectively detected live *Listeria monocytogenes* in pork. Lien [29] used the vPCR strategy to rapidly and quantitatively detect live *Helicobacter pylori* in 50 retail pork samples and evaluated its survival time in pork (at least 48 h). The concentrations of live bacteria detected in two samples by this method were 4 bacteria/g and 49 bacteria/g, respectively. The problem that the culture-based detection method cannot successfully detect the viability of *H. pylori* in food due to the low success rate of isolation and culture of *H. pylori* was successfully solved.

In summary, RT-PCR and vPCR, can effectively detect live bacteria in each link of the food production chain and monitor the survival status of bacteria in food, thereby reducing the risk of human infection with foodborne diseases.

## 5. Summary and Outline

Effectively distinguishing live and dead bacteria is of grave importance in the process of food-borne pathogens detection. It will help food regulatory authorities adequately assess the growth of pathogenic bacteria in food, provide timely control of the food safety risks, and improve the healthy and sustainable development of the food industry. Various methods have been proposed to replace the traditional cell culture methods but have failed due to many obvious limitations. Now, the considerable progress in amplifying the detection signal and shortening the detection time have been made possible by the novel PCR methods. Therefore, we provide a unique perspective on PCR-based biochemical technology for viable and dead foodborne pathogens detection. Based on the different distinguishment principles (the intervention of DNA/RNA probes, specific phages, and aptamers), as well as the powerful assistance of PCR in the signal amplification process after recognition, modern viable and dead foodborne pathogens detection technology has been developing rapidly. For the DNA/RNA-based PCR detection technique, the better universality and stronger operability are beneficial for gaining extensive applications in bacteria identification.

There is no doubt that PCR paves the way for viable and dead foodborne pathogens detection, although various isothermal and non-enzymatic nucleic acid amplification technologies are constantly being created. With excellent versatility and stable amplification ability, the position of PCR in food safety remains unassailable. Nevertheless, the emerging amplification technologies are still expected to broaden application scenarios with convenient detection operation, and intensive signal output.

## Figures and Tables

**Figure 1 foods-11-02675-f001:**
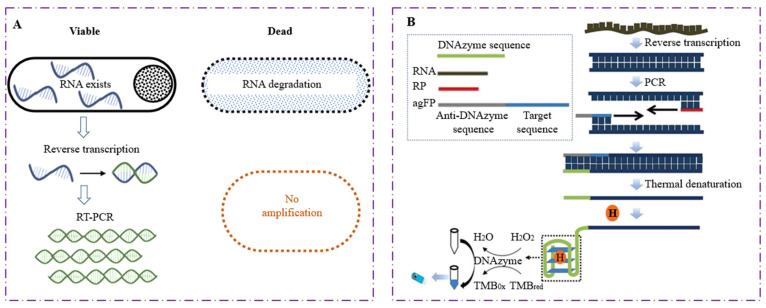
Principle diagram of viable bacteria detection based on RT-PCR. (**A**) Schematic diagram conventional RT-PCR amplification; (**B**) Schematic diagram of electrochemical biosensor for detection of *Cronobacter sakazakii* by RT-PCR-triggered G-quadruplex DNA enzyme catalytic reaction [8].

**Figure 2 foods-11-02675-f002:**
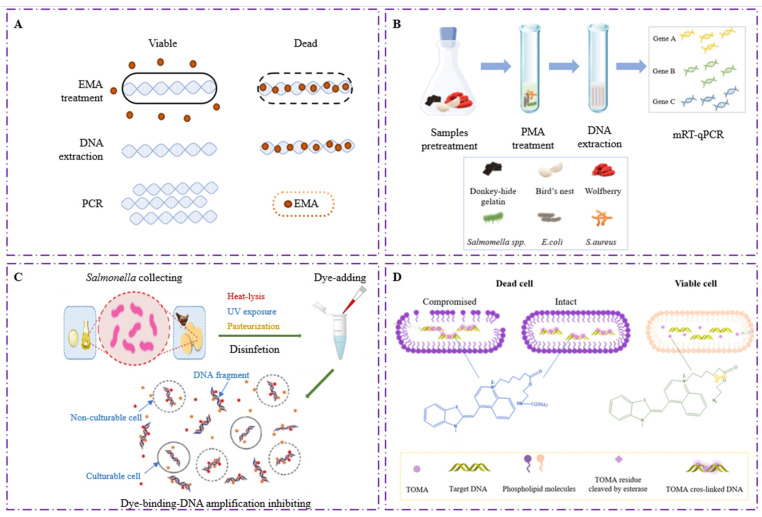
Schematic diagram of four nucleic acid intercalation dyes combined with PCR for distinguishing and detecting viable and dead bacteria. (**A**) Principal diagram of EMA-PCR for detecting viable and dead bacteria based on cell membrane integrity [10]; (**B**) Flow chart of multiple real-time detection of viable bacteria by PMA-PCR [14]; (**C**) Flow chart of DyeTox13-PCR detection of viable bacteria in sterilized samples [16]; (**D**) Principal diagram of TOMA-PCR for detecting viable and dead bacteria based on enzyme activity [18].

**Figure 3 foods-11-02675-f003:**
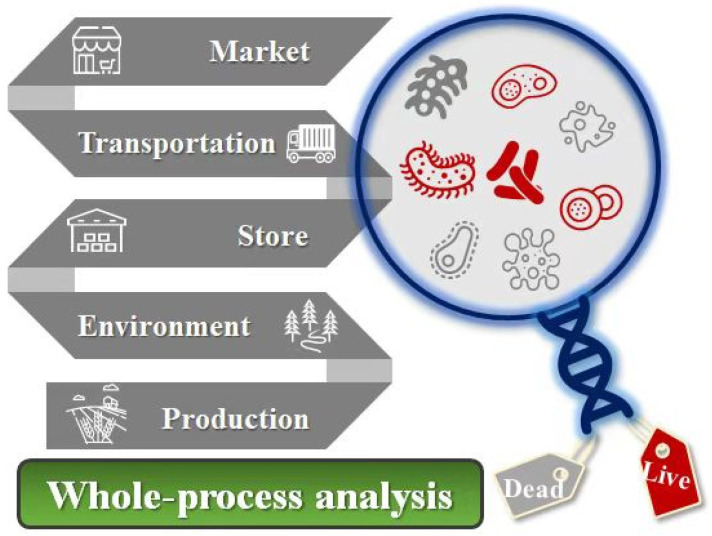
Graphics of the whole process of PCR-mediated nucleic acid molecular recognition technology applied to the detection of foodborne pathogens.

**Table 1 foods-11-02675-t001:** Comparison of different PCR techniques for detecting viable foodborne pathogens.

Type	Material	Time	Target	LOD	Effect	References
RNA-based RT-PCR	RNA reverse transcriptase	2016	*Mycobacterium tuberculosis*	-	Shorter detection time, fewer detection steps and easier operation	[6]
		2020	*Cronobacter sakazakii*	501 CFU/mL	Dynamic range was 2.4 × 10^7^ CFU/mL–3.84 × 10^4^ CFU/mL	[8]
		2021	*Pseudomonas aeruginosa*	1 cell/mL in blood and 100 cells/g in stool	The method can directly and rapidly quantify *PA* in clinical samples within 6 h without cross-reaction	[19]
DNA-based vPCR	EMA	2017	*Escherichia coli*	-	Reduced sensitivity when detecting UV-treated samples	[20]
		2021	*Legionella* *pneumophila*	-	EMA has no dye toxicity to VBNC bacteria	[21]
	PMA	2021	*Salmonella*	100 per gram of soil	High specificity (92%)	[22]
		2022	*Salmonella* spp., *Escherichia coli*, and *Staphylococcus aureus*	*Salmonella*:100 *E. coli*:100 *S. aureus*:10 CFU/mL	Multiplex detection	[13]
	DyeTox13	2018	*P. aeruginosa PAO1* and *Enterococcus faecalis v583*	-	Accurate assessment of the survival status of both Gram-negative and Gram-positive bacteria	[14]
		2022	*Salmonella typhimurium*	-	It accurately detects the number of viable bacteria in UV-sterilized samples	[15]
	TOMA	2019	*Escherichia coli*	1000 CFU/mL	It can work in the extreme conditions such as strong radiation	[16]
		2022	*Klebsiella pneumoniae*	2.3 × 10^4^ CFU/mL	It can be completed within 40 min at a constant temperature	[17]

## Data Availability

Data is contained within the article.

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
