# Peer review of "PCR Mediated Nucleic Acid Molecular Recognition Technology for Detection of Viable and Dead Foodborne Pathogens"

_foods, 2022, doi:10.3390/foods11172675_

Round 1

Reviewer 1 Report

Authors described the PCR mediated nucleic acid molecular recognition technology for viable and dead foodborne pathogens detection. Generally, this review provides a useful reference for the development of detection of foodborne pathogens.

However, the authors added that each method should be further investigated to which bacteria and which foods it was mainly applied to and added relevant references for each method. 

1. The reference form cited in the text needs to be modified to fit the Foods format.

2. Line 16: including RNA-based reverse transcriptase PCR

3. line 17: in recent years. They can not only

4. Figure 2 A, C: Requires correction by DNA extraction

5. lines 59-60: Change dependant to dependent 6. line 75: change equipments to equipment or types of equipment

7. line 100: change Fusarium sakazakii to Cronobacter sakazakii

8. line 135: delete “viable but nonculturable” and change to VBNC cells

9. lines 144-145: change gram-negative to Gram-negative, change gram-positive to Gram-positive.

10. line 145: change E. faecalis v583 to Enterococcus faecalis v583

11. In Sections 2 and 3, it is necessary to add a paragraph on the limitations and future directions of RNA-based RT-PCR and DNA-based vPCR developed so far, respectively.

12. In Section 3, additional paragraphs on the EMA-PCR/PMA-PCR, DyeTox13-PCR/TOMA-PCR methods are divided into 3.1 and 3.2 to summarize the advantages and disadvantages of each method.

Author Response

Response to Reviewer 1 Comments

Thanks for all the reviewer’s valuable comments. The following is a point-by-point response to the reviewers' comments.

Reviewer Comments to Author:

Authors described the PCR mediated nucleic acid molecular recognition technology for viable and dead foodborne pathogens detection. Generally, this review provides a useful reference for the development of detection of foodborne pathogens. However, the authors added that each method should be further investigated to which bacteria and which foods it was mainly applied to and added relevant references for each method.

 Response:

   Thanks for your kind comments and valuable suggestions. We have supplemented a table in line 202-203 about the list of reference observation PCR-technology with specified parameters, including detection limit, working range, and pathogen species. The detail revision is as follow:

Type

Material

Time

Target

LOD

Effect

References

RNA-based RT-PCR

RNA reverse transcriptase

2016

Mycobacterim tuberculosis

__

Shorter detection time, fewer detection steps and easier operation

[6]

2020

Cronobacter sakazakii

501 CFU/mL

Dynamic range was 2.4 × 107 CFU/mL - 3.84 × 104 CFU/mL

[8]

2021

Pseudomonas aeruginosa

1 cell/mL in blood and 100 cells/g in stool

The method can directly and rapidly quantify PA in clinical samples within 6 hours without cross-reaction

[26]

DNA-based vPCR

EMA

2017

Escherichia coli

__

Reduced sensitivity when detecting UV-treated samples

[27]

2021

Legionella

pneumophila

__

EMA has no dye toxicity to VBNC bacteria

[28]

PMA

2021

Salmonella

100 per gram of soil

High specificity (92 %)

[29]

2022

Salmonella spp., Escherichia coli, and Staphylococcus aureus

Salmonella:100

E. coli:100

S. aureus:10 CFU/mL

Multiplex detection

[14]

DyeTox13

2018

P. aeruginosa PAO1 and Enterococcus faecalis v583

__

Accurate assessment of the survival status of both Gram-negative and Gram-positive bacteria

[15]

2022

Salmonella Typhimurium

__

It accurately detects the number of viable bacteria in UV-sterilized samples

[16]

TOMA

2019

Escherichia coli

1000 CFU / mL

It can work in the extreme conditions such as

strong radiation

[17]

2022

Klebsiella pneumoniae

2.3×104 CFU/mL

It can completed within 40 minutes at a constant temperature

[18]

Table 1. Comparison of different PCR techniques for detecting viable foodborne pathogens.

Question 1- The reference form cited in the text needs to be modified to fit the Foods format.

Response:

Many thanks for your kind suggestion. We have modified the citation format according to the format required by Foods: cancel the upper corner and lower line. The detail revision is as follow:

“In addition to spores, there are three living states of foodborne pathogens: viable, dead and viable but nonculturable (VBNC) cells [1].” was revised as “In addition to spores, there are three living states of foodborne pathogens: viable, dead and viable but nonculturable (VBNC) cells [1].”

  1. Cangelosi, G. A.; Meschke, J. S. Dead or Alive: Molecular assessment of microbial viability. Applied and Environmental Microbiology. 2014, 80(19), 5884–5891.

Question 2- Line 16: including RNA-based reverse transcriptase PCR.

Response:

Many thanks for your advice. We have inserted “RNA-based” in line 16 to perfect the description of reverse transcriptase PCR to distinguish it from DNA-based PCR. The detail revision is as follow:

“reverse transcriptase PCR (RT-PCR), DNA-based viability PCR (vPCR)” was revised as “RNA-based reverse transcriptase PCR (RT-PCR), DNA-based viability PCR (vPCR)”

Question 3- line 17: in recent years. They can not only

Response:

Many thanks you for paying attention to our work. We feel sorry for the sentence lacking of punctuation and have inserted “.”in line 17 to split the two sentences. The detail revision is as follow:

“have been developed in recent yearsThey can not only” was revised as “have been developed in recent years. They can not only”

Question 4- Figure 2 A, C: Requires correction by DNA extraction

Response:

Many thanks for your kind suggestion. We apologize for the spelling error and correct the words in Figure 2 A, B to “DNA extraction”. The detail revision is as follow:

Question 5- lines 59-60: Change dependant to dependent

Response:

Thanks for your suggestion. We have modified the words in line 59-60. The detail revision is as follow:

“is almost entirely dependant on the isolation” was revised as “is almost entirely dependent on the isolation”

Question 6- line 75: change equipments to equipment or types of equipment

Response:

Thanks for your suggestion. We have modified the words in line 75. The detail revision is as follow:

“with basic molecular biology equipments,” was revised as “with basic molecular biology equipment,”

Question 7- line 100: change Fusarium sakazakii to Cronobacter sakazakii

Response:

Thanks for your suggestion. After checking the citations, we modified the name of the foodborne pathogen in line 106. The detail revision is as follow:

“for detection of Fusarium sakazakii by RT-PCR-triggered” was revised as “for detection of Cronobacter sakazakii by RT-PCR-triggered”

Question 8- line 135: delete “viable but nonculturable” and change to VBNC cells

Response:

Thanks for your suggestion. We have deleted the miscellaneous description of line 144. The detail revision is as follow:

“However, for “viable but nonculturable” (VBNC cells) that may exist in UV-treated samples,” was revised as “However, for VBNC cells that may exist in UV-treated samples,”

Question 9- lines 144-145: change gram-negative to Gram-negative, change gram-positive to Gram-positive

Response:

Thanks for your suggestion. We changed the first letter of “gram” in line 162-163 to uppercase format. The detail revision is as follow:

“distinguish between active/inactive gram-negative bacteria (P. aeruginosa PAO1) and gram-positive bacteria” was revised as “distinguish between active/inactive Gram-negative bacteria (P. aeruginosa PAO1) and Gram-positive bacteria”

Question 10- line 145: change E. faecalis v583 to Enterococcus faecalis v583

Response:

Thanks for your suggestion. We have supplemented the abbreviation of bacterial name in line 163-164. The detail revision is as follow:

“and Gram-positive bacteria (E. faecalis v583).” was revised as “and Gram-positive bacteria (Enterococcus faecalis v583).”

Question 11- In Sections 2 and 3, it is necessary to add a paragraph on the limitations and future directions of RNA-based RT-PCR and DNA-based vPCR developed so far, respectively.

Response:

Many thanks for your valuable advice. We have supplemented the limitations and future directions of RNA-based RT-PCR and DNA-based vPCR in line 94-103. The detail revision is as follow:

Delete “However, the half-life of mRNA is very short (1.5-2 min), and it is a challenge to purify and obtain complete mRNA. So that it is necessary to improve the RT-PCR scheme to develop a larger application platform in the detection of viable bacteria.”

Supplement “In summary, there are many advantages of RNA-based RT-PCR in the application of detecting viable bacteria , such as low sensitivity, strong specificity, and many emerging portable visualization schemes have been widely established and applied. However, the half-life of mRNA is very short (1.5-2 min), and it is a challenge to purify and obtain complete mRNA. Therefore, it is necessary to improve the RT-PCR scheme and optimize the pretreatment steps to develop a larger application platform in viable bacteria detection.”

Question 12- In Section 3, additional paragraphs on the EMA-PCR/PMA-PCR, DyeTox13-PCR/TOMA-PCR methods are divided into 3.1 and 3.2 to summarize the advantages and disadvantages of each method.

Response:

Many thanks for your constructive suggestions. According to the coloration principle of dyes, we divided the four dyes into two sections, and summarized the advantages and disadvantages of their detection applications. The detail revision is as follow:

Supplement the heading of section in line 109 “3.1 Identification of live and dead bacteria based on cell membrane integrity ”

Supplement in line 140-148 “As the earliest DNA-intercalating dye, EMA is cheaper and more widely used than PMA. At the same time, PMA is more used by researchers to combine with multiplex PCR to achieve high-throughput detection. Overall, EMA-PCR and PMA-PCR have been successfully applied to distinguish living and dead bacteria in various samples such as water, food, air and clinical specimens. However, for  VBNC cells that may exist in UV-treated samples, above two methods may cause increased false positive results that overestimate the number of viable cells in samples. Therefore, it is necessary to further improve the accuracy of EMA-PCR and PMA-PCR in distinguishing and detecting viable bacteria in future applications.”

In line 149-155 Supplement the heading of section “3.2 Identification of living and dead bacteria based on cell metabolic activity” and “EMA-PCR and PMA-PCR have been successfully applied to distinguish living and dead bacteria in various samples such as water, food, air and clinical specimens. However, for “viable but nonculturable” (VBNC) cells that may exist in UV-treated samples, above two methods may cause increased false positive results that overestimate the number of viable cells in samples. To make up for this shortcoming,”  was revised as “In order to improve the above defects of high false positive rate of vPCR,”

Supplement in 187-190 “As the new nucleic acid intercalation dye, DyeTox13 and TOMA can correctly distinguish the survival state of bacteria after various sterilization conditions, and do not require additional fluorescent dyes or probes, which can achieve the effect of visual detection by itself without additional fluorescent dyes or probes themselves.”

Reviewer 2 Report

Review of “PCR mediated nucleic acid molecular recognition technology for viable and dead foodborne pathogens detection” by Chen et al.
Review manuscript "PCR mediated nucleic acid molecular recognition technology for viable and dead foodborne pathogens detection" by Chen et al. is devoted to summarizing recently developed methods for the detection of foodborne pathogens at the molecular level, based on polymerase chain reaction. In addition, a brief description of the whole-processes application of the listed PCR mediated methods is given. Several techniques are observed with different sensitivities, readoutformats, and discrimination between live/dead and non-culturable pathogens. It should be noted that the text is written clearly and has a significant practical output. The total volume of the manuscript text is rather small, which gives the reader only a superficial idea of the problem. The list of references includes only 25 cited sources, and only half of them have been published in the last 5 years. Therefore, I should address several major points to the authors:
1) To better systemize the given material, an addition of table is highly recommended including the list of the referred observed PCR-technologies with specified parameters, such as detection limit, working range, type of pathogens etc.
2) List of reference should be increased, with accent to novel technologies.
3) Just as suggestion to authors, is to add a special subsection for comparative discussion of visual (naked-eye) nucleic acids detection systems, or also called portable non-instrument techniques, their advantages and challenges in comparison to others PCR-based methods for pathogen discrimination

Author Response

Response to Reviewer 2 Comments

Thanks for all the reviewer’s valuable comments. The following is a point-by-point response to the reviewers' comments.

 Review manuscript "PCR mediated nucleic acid molecular recognition technology for viable and dead foodborne pathogens detection" by Chen et al. is devoted to summarizing recently developed methods for the detection of foodborne pathogens at the molecular level, based on polymerase chain reaction. In addition, a brief description of the whole-processes application of the listed PCR mediated methods is given. Several techniques are observed with different sensitivities, readoutformats, and discrimination between live/dead and non-culturable pathogens. It should be noted that the text is written clearly and has a significant practical output. The total volume of the manuscript text is rather small, which gives the reader only a superficial idea of the problem. The list of references includes only 25 cited sources, and only half of them have been published in the last 5 years.

 Response:

   Thanks for your valuable comments and questions, we have added x4 papers in line 349-357 published in recent five years that used PCR to detect live foodborne pathogens. The detail revision is as follow:

  1. Mai N.; Satomi A.; Akira T.; Yukiko K.; Takuya S.; Hirokazu T.; Kentaro S.; Hiroshi O.; Takashi A. Development of a rapid and sensitive analytical system for Pseudomonas aeruginosa based on reverse transcription quantitative PCR targeting of rRNA molecules. Emerging Microbes & Infections. (2021), 10:1, 677-686.
  2. Yan R.; Liu Y.; Gurtler J.; Killinger K; Fan X. Sensitivity of pathogenic and attenuated coli O157:H7 strains to ultraviolet-C light as assessed by conventional plating methods and ethidium monoazide-PCR. Journal of Food Safety. 2017, 37(4), e12346.
  3. Michela C.; Anna Grassi.; Marina T.; Stefania S.; Maria Gori.; Elisabetta T. Assessing the viability of Legionella pneumophila in environmental samples: regarding the filter application of Ethidium Monoazide Bromide. Annals of Microbiology. 2021, 71(1).

 Zhang J.; Khan S.; Chousalkar K. Development of PMAxxTM-Based qPCR for the quantification of viable and non-viable load of Salmonella from poultry environment. Frontiers in Microbiology. 2020,11:581201.

Question 1- To better systemize the given material, an addition of table is highly recommended including the list of the referred observed PCR-technologies with specified parameters, such as detection limit, working range, type of pathogens etc.

Response:

Thanks for your kind comments and valuable suggestions. We have supplemented a table in line 202-203 about the list of reference observation PCR-technology with specified parameters, including detection limit, working range, and pathogen species. The detail revision is as follow:

Type

Material

Time

Target

LOD

Effect

References

RNA-based RT-PCR

RNA reverse transcriptase

2016

Mycobacterim tuberculosis

__

Shorter detection time, fewer detection steps and easier operation

[6]

2020

Cronobacter sakazakii

501 CFU/mL

Dynamic range was 2.4 × 107 CFU/mL - 3.84 × 104 CFU/mL

[8]

2021

Pseudomonas aeruginosa

1 cell/mL in blood and 100 cells/g in stool

The method can directly and rapidly quantify PA in clinical samples within 6 hours without cross-reaction

[26]

DNA-based vPCR

EMA

2017

Escherichia coli

__

Reduced sensitivity when detecting UV-treated samples

[27]

2021

Legionella

pneumophila

__

EMA has no dye toxicity to VBNC bacteria

[28]

PMA

2021

Salmonella

100 per gram of soil

High specificity (92 %)

[29]

2022

Salmonella spp., Escherichia coli, and Staphylococcus aureus

Salmonella:100

E. coli:100

S. aureus:10 CFU/mL

Multiplex detection

[14]

DyeTox13

2018

P. aeruginosa PAO1 and Enterococcus faecalis v583

__

Accurate assessment of the survival status of both Gram-negative and Gram-positive bacteria

[15]

2022

Salmonella Typhimurium

__

It accurately detects the number of viable bacteria in UV-sterilized samples

[16]

TOMA

2019

Escherichia coli

1000 CFU / mL

It can work in the extreme conditions such as

strong radiation

[17]

2022

Klebsiella pneumoniae

2.3×104 CFU/mL

It can completed within 40 minutes at a constant temperature

[18]

Table 1. Comparison of different PCR techniques for detecting viable foodborne pathogens.

Question 2-  List of reference should be increased, with accent to novel technologies.

Response:

Thanks for your valuable comments and questions. We added4 papers in line 349-357 published in recent five years that used PCR to detect live foodborne pathogens. The detail revision is as follow:

  1. ai N.; Satomi A.; Akira T.; Yukiko K.; Takuya S.; Hirokazu T.; Kentaro S.; Hiroshi O.; Takashi A. Development of a rapid and sensitive analytical system for Pseudomonas aeruginosa based on reverse transcription quantitative PCR targeting of rRNA molecules. Emerging Microbes & Infections. (2021), 10:1, 677-686.
  2. Yan R.; Liu Y.; Gurtler J.; Killinger K; Fan X. Sensitivity of pathogenic and attenuated coli O157:H7 strains to ultraviolet-C light as assessed by conventional plating methods and ethidium monoazide-PCR. Journal of Food Safety. 2017, 37(4), e12346.
  3. Michela C.; Anna Grassi.; Marina T.; Stefania S.; Maria Gori.; Elisabetta T. Assessing the viability of Legionella pneumophila in environmental samples: regarding the filter application of Ethidium Monoazide Bromide. Annals of Microbiology. 2021, 71(1).
  4. Zhang J.; Khan S.; Chousalkar K. Development of PMAxxTM-Based qPCR for the quantification of viable and non-viable load of Salmonella from poultry environment. Frontiers in Microbiology. 2020,11:581201.

Question 3-  Just as suggestion to authors, is to add a special subsection for comparative discussion of visual (naked-eye) nucleic acids detection systems, or also called portable non-instrument techniques, their advantages and challenges in comparison to others PCR-based methods for pathogen discrimination.

Response:

Many thanks for your kind advice. In our manuscript, we focused on discussing the PCR-based nucleic acid molecular technology that can distinguish between live and dead bacteria. We divide paragraphs based on the identification of different target types (RNA/DNA) to help readers better overview from the technical principal level. Additionally, the emerging technology in line 83-85:(Yuan et. al [8] established a cascade signal amplification strategy based on RT-PCR triggering G-quadruplex DNA enzymatic reaction, realizing the visualization and rapid detection of viable Cronobacter sakazakii in samples) has been mentioned. Thus, we think the visual (naked eye) nucleic acid detection system has already been described as is relatively clear, and tend to hold on the original paragraphs form of our manuscript. Thanks again for your attention to our work’s quality improvement.

Round 2

Reviewer 1 Report

No additional comments.

Reviewer 2 Report

The authors provided point-by-point responces to my remarks, therefore the manuscript may be considered for publishing in Foods as is.